# Dialysis Membranes for Acute Kidney Injury

**DOI:** 10.3390/membranes12030325

**Published:** 2022-03-15

**Authors:** Yanuardi Raharjo, Muhammad Nidzhom Zainol Abidin, Ahmad Fauzi Ismail, Mochamad Zakki Fahmi, Muthia Elma, Djoko Santoso, Hamizah Haula’, Ahlan Riwahyu Habibi

**Affiliations:** 1Membrane Science and Technology Research Group (MSTRG), Chemistry Department, Faculty of Science and Technology, Universitas Airlangga, Surabaya 60115, Indonesia; m.zakki.fahmi@fst.unair.ac.id (M.Z.F.); hamizah.haula-2020@fst.unair.ac.id (H.H.); ahlan.riwahyu.habibi-2020@fst.unair.ac.id (A.R.H.); 2Department of Chemistry, Faculty of Science, Universiti Malaya, Jalan Profesor Diraja Ungku Aziz, Kuala Lumpur 50603, Malaysia; nidzhom@um.edu.my; 3Advanced Membrane Technology Research Centre (AMTEC), Universiti Teknologi Malaysia, Skudai 81310, Malaysia; afauzi@utm.my; 4Chemistry Department, Faculty of Mathematics and Natural Science, Syiah Kuala University, Banda Aceh 23111, Indonesia; saiful@unsyiah.ac.id; 5Chemical Engineering Department, Lambung Mangkurat University, Jl. A. Yani KM 36, Banjarbaru 70123, Indonesia; melma@ulm.ac.id; 6Division of Nephrology and Hypertension, Dr. Soetomo Hospital, Faculty of Medicine, Universitas Airlangga, Surabaya 60115, Indonesia; drdjokosantoso@yahoo.com

**Keywords:** acute kidney injury, haemodialysis membrane, mixed matrix membrane, haemoperfusion, adsorption

## Abstract

Mortality and morbidity rates among critically ill septic patients having acute kidney injury (AKI) are very high, considering the total number of deaths after their admission. Inappropriate selection of the type of continuous renal replacement therapy and inadequate therapy become the immediate causes of these issues. Dialysis is a commonly used treatment intended to prolong the life of AKI patients. Dialysis membranes, which are the core of dialysis treatment, must be properly selected to ensure fair treatment to the patients. The accumulation of certain types of molecules must be dealt with using the right membrane. Whether it is low-flux, high-flux, or adsorptive type, the dialysis membrane should be chosen depending on the condition of the patients. The selection of dialysis membranes should also be based on their effect on the treatment outcomes and well-being. All these options are needed to serve the patients of different clinical settings. The use of dialysis membranes is not restricted to conventional haemodialysis, but rather they can be employed in haemoperfusion, haemofiltration, haemodiafiltration, or a combination of any two of them. This review focuses in-depth on different types of dialysis membranes, their characteristics, and approaches in addressing the issues encountered in patients having AKI with sepsis and/or multiorgan failure in intensive care units.

## 1. Introduction

Membrane technology is growing rapidly. At the beginning of the use of this technology, membranes were widely applied to support the development of science and technology in theoretical physics and chemistry, rather than for commercial use. Due to various technological benefits, researchers began developing membrane applications for human survival. One of the applications is for kidney disease treatment, namely haemodialysis (HD) through a principle of dialysis, whereby the membrane acts as an artificial kidney. Haemodialysis is not intended to heal the patient but to prolong the life of the patient in acute and chronic conditions. Acute kidney injury (AKI) and acute-on-chronic kidney failure (ACKF) are conditions when the patients experience a sudden negative change in blood quality caused by the accumulation of toxins.

Specifically, AKI is a condition in which the concentration of creatinine in serum increases >50% more than its normal concentration in a very short time (<7 days). The urine output becomes less than 0.5 mL/kg/h for more than 6 h. Based on the National Kidney Foundation (NKF) in the USA, around 37 million people have been affected by kidney disease, affecting 15% of the adult population. There are different conditions of AKI, namely prerenal, renal, and postrenal. In prerenal and postrenal conditions, the blood flow to the kidney and the urine flow from the kidney is affected. Meanwhile, the renal condition can be caused by glomerulonephritis, blood clotting, and blood vessel disease, whereby the kidney fails to purify the blood. Patients with AKI and ACKF have to be treated using HD [1].

Acute Dialysis Quality Initiative has classified AKI based on the consensual risk, injury, failure, loss of kidney function, and end-stage kidney disease (RIFLE). The RIFLE classification is based on serum creatinine (sCr) and urine output (UO). Based on the glomerular filtration rate, the risk, injury, and failure are occurring while the sCr increases 1.5, 2, and 3 times, respectively. The patients are defined to have the loss of kidney function and the end-stage kidney disease when their kidney function has been lost for more than 4 weeks and more than 3 months, respectively. Based on the UO, the risk, injury, and failure are classified with less than 0.5 mL/kg/h for 6 h, less than 0.5 mL/kg/h for 12 h, and less than 0.3 mL/kg/h for 24 h or anuria for 12 h, respectively. The RIFLE is used as a guideline to diagnose and justify the patient’s condition for kidney conditions [2]. In March 2012, the National Kidney Foundation–Kidney Disease Outcomes Quality Initiative completed the AKI regulation for adults and paediatrics [3].

There are two dialysis treatments for AKI patients, namely intermittent haemodialysis (IHD) and continuous renal replacement therapy (CRRT). The main focus of the dialysis treatments in AKI is to remove excess water and waste. IHD is used for a short period of time (3–4 h), while CRRT is conducted continuously (24 h) for several days. CRRT is applicable for critically ill patients having different catabolic states, systemic inflammatory syndromes with or without sepsis, and other organ failures. CRRT does not cause abrupt variations in fluid removal or osmolality, ensuring good clearance of solute and better haemodynamic tolerance due to the slower liquid flow rate [4].

A kidney is a vital organ to clean the body fluid from acidic, organic, and metabolic waste through a series of urine production stages that cover water and toxin clearance. Kidney failure is one of the significant health problems of the world population suffering from the disease. It refers to the incapability of kidneys to perform their essential tasks: eliminating waste products from body metabolisms (i.e., urea, creatinine, and excess water) and maintaining electrolyte balance in the body. It is commonly caused by certain conditions, such as diseases (i.e., diabetes, hypertension) and injuries that induce sepsis or systemic inflammatory response syndrome. AKI is a sudden reduction of kidney function within 48 h, indicated by the increasing concentration of creatinine in serum of equal to or more than 0.3 mg/dL, the increasing percentage of creatinine of equal to or more than 50%, or the reduction of excreted urine of less than 0.5 mL/kg per hour for more than 6 h [5]. AKI can be detected by symptoms like pallor, leukonychia, pulmonary oedema, raised blood pressure, peripheral oedema, pleural effusion, tiredness, loin pain, anorexia, itching, nausea, vomiting, and haematuria. Hence, a patient with AKI is required to receive immediate treatment from the doctor to keep him alive.

Haemofiltration (HF), haemodialysis (HD), and haemodiafiltration (HDF) are among the treatment options for chronic kidney failure conditions like AKI and ACKF. The selection of dialysis membrane is very important to achieve adequate treatment. The patients have to be assessed in terms of their quality of blood and the amount of uremic toxins present prior to HD. Failure to choose the suitable dialysis membrane may cause an adverse effect and worsen the patient’s condition.

Uremic toxins are defined as the products of metabolism that accumulate in the body and their accumulation is associated with uraemia due to renal degradation and/or impaired excretory capacity. Based on the physicochemical characteristics, uremic toxins are generally divided into three groups: small water-soluble uremic toxins (molecular weight <500 Da), middle molecules (molecular weight range of 300–12,000 Da), and protein-bound uremic toxins [6]. The accumulation of each type of uremic toxins can also affect every human body system.

Vanholder et al. [7] stated that the adequacy of dialysis is evaluated by urea clearance in terms of blood urea nitrogen (BUN). Normally, the patients need a longer dialysis time (4.5 h) to remove 50% of urea from the blood. They found that achieving a lower BUN level is more effective in improving the treatment outcome rather than increasing the session duration. This can be achieved by the proper selection of dialysis membrane and treatment. During dialysis, small water-soluble uremic toxins are usually removed by simple diffusion, middle molecules are removed with the help of ultrafiltration (UF) or convection, whereas protein-bound uremic toxins are removed with the help of adsorption (haemoperfusion) treatment.

## 2. Uremic Toxins

### 2.1. Small Water-Soluble Uremic Toxins

Small water-soluble uremic toxins are easily removed by HD [8]. These uremic toxins dissolve in water thus are easily carried away by the dialysate solution via HD membranes, especially low-flux membranes in conventional dialysis. They are less than 500 Da, which includes urea, creatinine, and uric acid. Urea and creatinine are usually used as markers of successful dialysis. The small water-soluble uremic toxins can be subdivided into several groups like ribonucleosides, guanidines, purines, pyrimidines, and polyols, as shown in Table 1.

### 2.2. Middle Molecules

The middle molecules can be normally removed by dialysis using high-flux membranes, and the molecules are hardly cleared by conventional dialysis or low-flux membranes [9]. The middle molecules can be subdivided into peptides and cytokines groups (Table 1). The accumulation of β_2_-microglobulin would induce cardiovascular events, including stroke, myocardial infarction, and heart failure, until cardiovascular mortality in every stage of chronic kidney failure patients. These uremic toxins can be removed using high-flux membrane, adsorptive membrane, or by haemoperfusion. The middle-molecule uremic toxins are dangerous if accumulated in the blood as they may cause endothelial or leukocyte dysfunction and may exert proinflammatory and hepatotoxic effects that contribute to the increase of mortality by destroying six vital organs as kidneys, heart, lungs, liver, brain, and gastrointestinal.

### 2.3. Protein-Bound Uremic Toxins

The protein-bound uremic toxins can increase the body’s susceptibility to infection and cardiovascular complications. According to Sakai [10], there are a total of 25 protein-bound uremic toxins, which can be divided into phenols, hippurates, indoles, peptides, polyamines, and advanced glycation end-products (AGE) groups (Table 1). These uremic toxins can cause cardiorenal syndrome. Although they are small-sized molecules, their tendency to bind with proteins has made them difficult to be removed using a conventional dialyser. Hence, these toxins can be effectively removed using either haemoperfusion or HDF, which combines dialysis and adsorption mechanisms.

## 3. Dialysis Membrane

Haemodialysis is a replacement therapy for kidney failure patients to remove excess metabolic waste or uremic toxins from blood, such as water, sodium, potassium, hydrogen, urea, creatinine, uric acid, and other substances through a semi-permeable membrane, where the processes like diffusion, osmosis, and UF occur without losing the essential substances, such as glucose, electrolytes, and amino acids [11]. The principle of HD involves the movements of solutes and water from blood across the membrane into the dialysate. Large molecules (i.e., blood cells and proteins) are retained inside the blood. In contrast, smaller molecules (i.e., urea, creatinine, and other biological waste) will pass through the small pores of the membrane into the dialysate. Diffusion and UF are two fundamental processes that ensure continuous renal therapy. Diffusion refers to the movement of typically small solutes from a compartment in which they are in high concentration to another compartment in which they are in low concentration. Meanwhile, UF is a process whereby water molecules (blood plasma) are forced to move across a semi-permeable membrane by a pressure gradient. The rate of UF depends on the porosity of the membrane and the hydrostatic pressure of the blood, which depends on blood flow [12].

There are two compartments in the dialyser to contain blood and dialysate, separated by membranes. Dialysate flows in counter-current to the blood flow on the other side of the membrane to maximise solute concentration gradient for efficient diffusion. Diffusive clearance of a solute depends on its molecular weight and electrical charge, as well as the blood-dialysate concentration gradient, blood and dialysate flow rates, and the membrane characteristics in terms of diffusion coefficient [13]. Small molecules such as urea will move easily along the concentration gradient into the dialysate most of the time. However, more significant- or middle-sized molecules, which are believed to cause uraemia, are hardly removed by this process. Meanwhile, UF or convection is very effective for the removal of fluid along with middle-sized molecules from the blood into the dialysate and across the membrane.

Since the first creation of HD membrane, this membrane has been derived from polymeric materials. Membrane separation technology is effective because it is flexible, requires little energy, does not alter the molecular structure of the separated substance, can be operated at room temperature, and does not require additional chemical substances during separation [14]. The dialysis membrane must possess a selective transport property that can withstand larger species and skip smaller species through the membrane.

Several properties are required for a dialysis membrane, such as high solute permeability, high water permeability, the balance between solute and water permeability, mechanical strength in wet state, satisfactory biocompatibility, and low cost. HD membranes may become wet through contact with blood, where there will be a change in inner diameter, thickness, or length, and the membranes must also have excellent mechanical strength. The best membranes applied in HD are membranes with a large pore area, strong, stretchable, thin, and lightweight [6].

In general, membranes used for HD application are hollow fibre rather than flat sheet membranes. This is due to many drawbacks of a flat sheet membrane, such as frequent fouling, resulting in reduced membrane performance. In addition, the membrane also has a relatively small surface area when applied in HD. Compared to the flat sheet membrane, a hollow fibre membrane offers several advantages for HD. The performance of hollow fibre membrane is better due to its higher total surface area [15]. The surface area of the hollow fibre membrane has a surface area density of 3000 m^2^/m^3^ compared to a flat sheet membrane with a surface area density of 400 m^2^/m^3^. A hollow fibre membrane also has a stronger mechanical structure than a flat sheet membrane. However, to produce a good membrane, the hollow fibre must be thin with a very small diameter (about 200 nm) so that more toxins in the body can be eliminated from human blood [16].

A cellulose-based membrane without modification was used in the earlier development of HD membrane. This membrane is homogeneous, symmetric, and has good performance for small water-soluble molecules. In the 1970s, a synthetic membrane dialyser was developed with higher water permeability for the purpose of blood filtration. The most noticeable changes of this development are larger pore size, a thicker wall structure, higher hydrophobicity, more uniform pore size and distribution, and a more asymmetric membrane structure. Those changes influenced the performance of HD membranes to become better and more stable over a longer period [17]. Various synthetic membranes have been used clinically for HD, which are usually referred to as a single polymer name, including polysulfone (PSf), polyethersulfone (PES), polyamide, poly(aryl ether sulfone), polycarbonate, polyacrylonitrile, polymethylmethacrylate, and poly(ethylene-co-vinyl alcohol).

Hoenich [13] reviewed the fabrication of polyvinylidene fluoride (PVDF) hollow fibre membranes via the non-solvent induction phase method to be used for HD. Their study indicated that the hollow fibre membranes made of PVDF have advantages in mechanical properties and water flux. However, the membrane did not have anticoagulant properties and was easily fouled by protein. In other research, poly (lactic acid) (PLA) was used as the base material of HD membranes. PLA is one of the eco-friendly bioplastic materials that are easily broken, easily processed, and cheap. Many studies in the field of health materials used PLA. It is a polymer with hydrophilic properties and low electrical conductivity. During dialysis, pollutants in the body (e.g., proteins and microorganisms) are easily adsorbed and stored in the membrane matrix [18].

Meanwhile, PSf as a membrane is advantageous due to its high chemical resistance and is also not reactive in mineral acids, alkalis, and salts. Furthermore, PSf has excellent resistance over a wide temperature range (75–125 °C) and pH range (1–13). The main drawback is its higher fouling tendency than hydrophilic membranes [19]. Therefore, surface modifications are needed to solve the existing issues of hydrophobic membranes. Some materials can be added or blended to the membrane to solve this problem, such as using hydrophilic polymer, biomaterials, sorbent, and inorganic nanoparticles. Other options include surface coating and functionalisation.

Blending polymers is interpreted as a physical mixture that is not covalently bonded by accumulating the properties of different polymers into a single membrane [20]. This technique is the most widely used in the development of HD membranes, specifically to increase the hydrophilicity and biocompatibility of synthetic membranes. The optimum ratio of the hydrophilic materials blended to the hydrophobic polymer can be determined by the permeability and selectivity of the resultant membrane. The mixture of two or more materials can produce homogeneous (polymers miscible in all compositions) and heterogeneous (polymers not miscible in all compositions) membranes.

Researchers have carried out the blending of hydrophobic polymers like PSf and PES using a hydrophilic polymer like polyvinylpyrrolidone (PVP) [20,21]. A combination of both polymers may produce a patient-friendly membrane. PVP is a highly hydrophilic polymer without hydroxyl carbon and is non-ionic. PVP is known for its ability to inhibit protein adsorption on the membrane surface; therefore, it can increase the antifouling property and biocompatibility of the membrane [22,23]. The higher PVP loading in the dope composition increased the water flux and improved the biocompatibility of the membrane [24]. Other hydrophilising agents besides PVP are polyethylene glycol (PEG) and polypropylene glycol (PPG) that utilise water-soluble solvents, such as dimethylformamide (DMF), dimethylacetamide (DMAc), N-methylpyrrolidone (NMP), and dimethyl sulfoxide (DMSO) [25].

Biomaterials or biological compounds are also added to HD membranes as additives to increase membrane biocompatibility. Heparin [26] and vitamin E [27] have been successfully blended into hydrophobic polymers, such as PSf and PES. Other added materials can be used as adsorbents, like activated carbon (charcoal) [21], zeolite [28], and hydroxyapatite (HAP) [29]. Zeolites, on the other hand, are microporous, aluminosilicate minerals commonly used as commercial adsorbents and catalysts. Wernert et al. [26] successfully utilised zeolites in the development of HD membranes. The membrane could eliminate about 67% creatinine and 29% *p*-cresol. In addition, zeolite can be used and added to the HD membrane to clear middle-molecule toxins [27].

Recently, inorganic nanoparticles like carbon nanoparticles (carbon nanotube and graphene) and metal oxide nanoparticles (titanium dioxide and iron oxide) were used as nanofillers in HD membranes [28,29]. Nanoparticles are particles with a size between 1 and 100 nm. The application of these particles is based on their large surface area and water transport properties. They can increase membrane resistance towards chemical degradation, thermal stability, and fouling.

Similar to dialyzers, membranes can be categorised based on their flux and efficiency. A low-flux membrane is termed as having a UF coefficient of <10 mL/h/mmHg, whereas a high-flux membrane is a membrane with a UF coefficient of >20 mL/h/mmHg with middle-molecule (i.e., β_2_-microglobulin) clearance of >20 mL/min. On the other hand, membrane efficiency is determined based on the mass transfer area coefficient of urea, *KoA_urea_*. Membrane with KoA < 500 mL/min is called a low-efficiency membrane, whereas the one with KoA > 600 mL/min is known as a high-efficiency membrane. Although there are specific values to dictate which one is a high- or low-flux membrane, and whether it has high or low efficiency, membranologists often describe the flux and efficiency of the membranes in different terms (i.e., pure water flux and percentage removal of uremic toxins). Therefore, one could only tell which category the membrane belongs to based on the comparison with various developed membranes reported in research articles.

### 3.1. Low-Flux Membrane

Cellulosic membranes are usually denoted as low-flux membranes despite their pronounced hydrophilic nature. This is due to their symmetric structure and small mean pore size [2,14]. The uniform resistance acted upon the entire membrane wall makes cellulosic membranes suitable for diffusion of small water-soluble molecules, such as urea. The latter produced modified cellulosic membranes (i.e., cellulose acetate) with a larger mean pore size compared to the unmodified cellulosic membrane (22 µm). This resulted in higher porosity of the membranes. However, their weak hydraulic permeability still limits the separation performance [30]. They are incapable of extending their molecular weight range of solutes that can be removed. Their relatively denser structure is impermeable to middle-molecule uremic toxins. The low flux does not help the convection of large proteins passing through the membrane.

In addition, some unmodified synthetic membranes also display low permeability of middle molecules, such as proteins. PES and PSf membranes are prevalently employed for blood purification [20]. Despite the advantages of PES- and PSf-based membranes (e.g., excellent oxidative and hydrolytic stability and good mechanical properties [31]), their progress in HD application is always limited by their hydrophobic properties. Many studies have concluded that membrane fouling is directly related to hydrophobicity [32]. Membrane fouling consistently remains one of the greatest challenges to HD treatment. Fouling is caused by the deposition or adsorption of solutes like proteins on the membrane surface and into membrane pores [33]. This phenomenon subsequently reduces the membrane flux and disrupts the separation performance of the membrane.

Vilar and Farrington [34] concluded that low-flux membranes allow efficient diffusive removal of small molecular weight molecules like urea. Nevertheless, the membranes show poor convective removal of middle molecules. Despite the disadvantages of low-flux membranes, many studies have shown that the use of this type of membrane is still relevant to some patients if the dialysis adequacy (KT/V > 1.2) is fulfilled [35]. However, for patients having chronic kidney failure with a high accumulation of uremic toxins, high-flux membranes are necessary to achieve minimum adequacy of the dialysis treatment.

### 3.2. High-Flux Membrane

High-flux membranes are highly permeable and biocompatible, usually made up of synthetic polymers with certain modifications to increase the hydrophilicity of the membranes [36]. The high permeability of synthetic membranes is also contributed by their larger mean pore size that offers a higher UF rate at low pressure. The additional feature of this type of membrane over low-flux membranes is the enhanced convective removal of middle molecules while maintaining excellent removal of small molecular weight molecules via diffusion [37]. The main requirement for the HD setting in high-flux membranes is the use of a high-quality dialysate solution that is made up of ultrapure water with no detectable endotoxins (<0.03 endotoxin units per mL) to minimise chronic inflammation [38].

For PSf and PES membranes, the large amount of low molecular weight molecule removal is mainly because of the asymmetric structure and the higher UF coefficient, which are contributed by the larger pore size and higher porosity of the membranes [39]. Nevertheless, HD membranes must also be assessed in terms of their capacity to eliminate potentially deleterious middle molecules. Very few studies have successfully removed middle molecules efficiently. Yu et al. [40] produced a highly permeable thin-film nanofibrous composite membrane consisting of an ultrathin hydrophilic active layer of chemically cross-linked PVA and an electrospun PAN nanofibrous supporting layer. The membrane exhibited excellent selectivity by removing 45.8% of middle-molecular weight toxins. It also possessed good mechanical strength and comparable haemocompatibility.

In addition to improved middle-molecule clearance, a study by Mortada et al. [41] showed that high-flux PSf membranes are more superior than those of low-flux PSf membranes in terms of removing accumulated metals in the blood of kidney failure patients during HD, especially cadmium and lead. This is very helpful as cadmium and lead are toxic and can potentially induce adverse effects on the patients.

Despite their advantages, membranes with too high flux often trigger the loss of water content in blood among dialysis patients as excessive UF may deplete the blood volume. This exposes the patients to the risk of hypotension [34]. Therefore, the dialysis dose must be properly controlled as recommended by nephrologists. Kidney failure patients, especially those in high-risk groups, are suggested to use high-flux membranes to ensure their long survival [42]. Besides, studies show that high-flux membranes can minimise the occurrence of uremic syndromes, such as amyloidosis, dyslipidemia, polyneuropathy, and infection [43]. However, for the patients having an AKI with a high accumulation of uremic toxins in a short time, high-flux membranes are not necessary to achieve dialysis adequacy, whereas the patients with AKI have a problem with the haemodynamic system.

## 4. Haemofiltration and Haemodiafiltration

HF and HDF generally use high-flux membranes to remove larger-sized uremic toxins. High-flux membranes are known for high generation of hydrostatic pressure, in which uremic toxins are significantly removed together with plasma filtrate by convection. These membranes must ensure that the filtrate contains high concentrations of uremic toxins; otherwise, the treatment would not be efficient and might render hypertension. A plasma-like electrolyte solution made up of Type I water (ultrapure water) is used to replace the lost plasma filtrate.

Another concern is the susceptibility of essential proteins (i.e., albumin) to be removed due to the high-pressure gradient. Thus, the chosen membrane must possess a well-compromised flux (UF coefficient) to attain the intended selectivity. A slow and continuous HF or HDF is usually performed on patients with AKI [44]. Although most developed membranes are meant for HD, the reported data on their sieving properties could serve as useful information for the potential use in HF and HDF. Compared to HD, HF could achieve excellent middle-molecule clearance of up to 80%, while HDF that utilises diffusion and convection could remove both small and large uremic toxins more efficiently.

## 5. Adsorptive Membrane/Sorbent

Adsorption can be categorised as either chemisorption or physisorption. Bonding in chemisorption is stronger than physisorption as chemisorption is simply a chemical reaction with interactions between adsorbates and chemical bonding groups on the surface of adsorbents. In contrast, physisorption involves electrostatic interactions between adsorbates and the surface of adsorbents. Generally, the time needed to reach adsorption equilibrium is shorter for physisorption compared to chemisorption.

### 5.1. Sorbent in Haemodialysis

In HD, adsorption has been proposed as one of the alternative methods to eliminate uremic toxins, and most studies used adsorptive membranes to remove protein-bound toxins, which are hardly removed via diffusion and UF. Zeolite Y successfully removed *p*-cresol as protein-bounded uremic toxins both with and without modification [45]. Inspired by the concept of the adsorptive membrane, several attempts have been made to form a mixed matrix membrane (MMM) using the HDF principle. In general, MMM is a combination of organic and inorganic components comprised of the integration of fillers or adsorbents into the polymer matrix. The MMMs combine the selectivity of inorganic particles or sorbents with the high productivity of filtration membrane and have been applied to separate and recover proteins or enzymes. Based on the structure, MMM can be divided into two categories: dense and porous structures. A dense structured MMM is mostly used in gas separation, pervaporation, and fuel cell applications. Meanwhile, a porous structured MMM is designed for adsorption purposes. The fillers that can be incorporated into the MMM are nanomaterials, such as zeolite, carbon nanotube, metal-organic framework, charcoal, and others. These nanomaterials can also be functionalised in order to adopt the fillers to provide selectivity and absorptivity towards the targeted molecules [46]. Hence, there are two mechanisms (i.e., diffusion and adsorption) in one membrane module offered by MMM. Excess water can be removed by diffusion mechanism, and excess uremic toxins (cytokines) can be cleaned by the adsorption mechanism.

Tijink et al. [47] established a novel approach in blood cleaning by focusing on the improvement of the adsorption capacity of a membrane. They fabricated dual-layer (flat sheet) MMMs, which combined diffusion and adsorption in a single step. Activated carbon (AC) was incorporated in the mixture of PES and PVP. A dual-layer MMM was fabricated, in which a particle-free membrane layer was formed on top of an MMM layer containing AC. The dual-layer MMM had lower water permeability (~350 L/m^2^·h·bar) compared to that of the single-layer MMM (1800 350 L/m^2^·h·bar). The MMMs adsorbed 29 mg/g of AC at an equilibrium concentration of 0.05 mg/mL. When tested towards human plasma for 4 h, the MMMs were able to remove more than 80% of creatinine and para-aminohippuric acid, which is a protein-bound toxin.

A year after, Tijink et al. [48] used the same formulation to develop a dual-layer hollow fibre MMM with the same goal of removing creatinine and protein-bound toxins (i.e., hippuric acid (HA), indoxyl sulphate (IS), and *p*-cresyl sulphate (pCS)). The water permeability of the MMM was 58.4 350 L/m^2^·h·bar. Compared to the previous dual-layer flat sheet membrane, the dual-layer hollow fibre MMM developed in this study achieved a higher adsorption capacity of creatinine (100 mg/g of AC) at the equilibrium concentration of 0.05 mg/mL. The maximum adsorption capacity of creatinine, HA, and IS was 3064, 134, and 350 mg/g of AC, respectively. In the human blood plasma adsorption study, the MMM maintained 83% removal of creatinine, and it adsorbed 60% pCS, 90% IS, and 95% HA after 4 h of incubation. In the presence of albumin, IS and pCS were poorly removed by adsorption and diffusion even after 6 h of operation. In contrast, the amount adsorbed was higher for all protein-bound toxins during the convection experiment due to the pressure difference. Despite the better performance, albumin partially passed through the membrane during the convection experiment.

In 2016, another attempt was made by Pavlenko et al. [49]. They developed a low-flux dual-layer hollow fibre MMM with a smaller dimension. A commercial carbon-based sorbent (mesoporous Norit A Supra) with an average pore size of 3 nm was embedded in the MMM. The Norit A Supra-incorporated MMM showed promising results, where the adsorption capacity of creatinine was 2579 mg/m^2^ and the removal of IS and pCS was 30% and 125% better in comparison to the first generation MMM and up to 100% as compared to particle-free industrial membranes [48]. It was stated that the low UF coefficient of the MMM (3.35 mL/m^2^/h/mmHg) and a molecular weight-cut off around 12 kDa prevented albumin leakage while achieving excellent removal of protein-bound toxins.

Other previously used adsorbents in the membrane include hydroxy appetite (HAP). Although its application in HD is yet to be seen, HAP has good adsorption to protein and is usually used in the medical field due to its biocompatibility and bioactivity [29]. HAP can be incorporated into an HD membrane to improve the membrane’s adsorption capacity towards middle-sized proteins. In general, MMM is very beneficial for the removal of uremic toxins that cannot be removed by conventional dialyses, such as larger-molecular-weight and protein-bound uremic toxins. It offers an alternative way to overcome the limitations of diffusion coefficient, sieving properties, and flux of the membrane. The combination of adsorption and diffusion carries an advantage on HD applications. The incorporation of sorbents within a membrane is a very effective approach for efficient blood purification [50]. The integration of adsorbent into a dialysis membrane significantly improved the performance of the membrane. The development of this method is ensured to be safe as the cellular components of blood do not interact with the embedded sorbent [51]. This method revolutionises the previous method of adsorption, for instance, in blood detoxification techniques (e.g., haemoperfusion).

### 5.2. Haemoperfusion

Haemoperfusion is the adsorption of molecules onto the surface of a biocompatible adsorbent via direct contact in an extracorporeal circuit, as opposed to removing toxins and excess body fluids through a semi-permeable membrane [52]. The sorbent has to be sufficiently biocompatible to enable direct contact without the destruction of blood elements. Sorbents can come from synthetic or natural materials. The natural sorbents usually used are natural zeolite and carbons like charcoal and AC. Meanwhile, the synthetic sorbents can be obtained from the polymerisation of monomers, synthetic zeolite that can be modified to control the structure of the internal pore system, and ion-exchange resin [53]. Sorbents can be in granule form, powder, spheres, flakes, and cylindrical pellets. They are solid particles with a single particle diameter between 50 μm and 1.2 cm. The surface-area-to-volume ratio is extremely high in sorbent particles with a surface area varying from 300 to 1200 m^2^/g. The adsorbents are also defined as microporous (pore size < 20 Å), mesoporous (pore size between 20 and 500 Å), and macroporous (pore size > 500 Å). The sorbents must have high selectivity and good transport properties to catch up with the targeted analyte fast.

The US Food and Drugs Administration (FDA) has classified the haemoperfusion system into two classes of medical device: class II and class III. A device is considered a class II medical device when it is used for treating poison and drug overdoses. In contrast, a class III medical device is used to treat patients with liver deficiencies, such as in hepatic coma [54]. Based on the principles and benefits of haemoperfusion, it is possible to be applied for removing uremic toxins during HD.

The development of haemoperfusion for HD was continued using zeolites in adsorbing creatinine (representing the small water-soluble molecules) and IS (representing the protein-bound uremic toxins). It was reported that 0.025 g of 940-zeolite powders could eliminate 91% of 2 μmol creatinine in 5 min. Meanwhile, PES/zeolite membrane could adsorb 4948 μg creatinine per g membrane and 550 μg IS per g membrane. The adsorption mechanism was proposed to be via electrostatic attraction [55].

## 6. Roles in Removing Cytokines and Endotoxins

The presence of cytokines and endotoxins in the bloodstreams of patients with AKI and sepsis poses a threat to the patients’ life. In the development of haemoperfusion for HD, the researchers in [54] examined carbon nanomaterials in terms of their adsorption capabilities for blood cleansing. Non-porous carbon represented by graphene nanoplatelets (GNPs) and porous carbon polymorphs representing mesoporous carbon was compared to observe their adsorption capacity for cytokines. It was found that the adsorption kinetics of the non-porous carbon significantly outperformed mesoporous carbon. The GNPs could completely remove cytokines from the blood after 5 min. Moreover, the GNPs also maintained good performance when embedded into cryogels and polytetrafluoroethylene [54].

In recent years, a well-established sorption cartridge, namely CytoSorb (distributed by LINC Medical) is integrated with dialysis machines with the intent to adsorb cytokines. It is designed specifically to treat patients with sepsis. This new technology has shown excellent removal of a wide range of crucial cytokines that cannot be eliminated using existing blood purification processes [56]. The cartridge consists of porous polymer beads that capture cytokines depending on the level of cytokines in the blood. The higher the concentration of cytokines, the faster the adsorption rate. The cartridge is for single use but can withstand up to 7 days of continuous use.

A multicentre study conducted by Basu et al. [56] on 43 patients provided the most reliable clinical evidence of cytokine removal using CytoSorb. It was reported that CytoSorb significantly reduced the concentration of IL-6 (49.1%), IL-1ra (36.5%), and IL-8 (30.2%) in the patients’ blood. Single-centre clinical studies (<20 patients) performed in India, Italy, and Germany suggested that the patients with predicted high mortality could survive if they were given CytoSorb less than 24 h after admission. The high overall survival of 75% and 62.5% were reported, where the procalcitonin levels, sepsis-related organ failure assessment score, and simplified acute physiology score decreased. The reduction of the scores justified the positive impact of using CytoSorb on the haemodynamic parameters, which is important for patients with AKI and sepsis. In terms of safety, there were no serious device-related adverse effects observed during the treatment.

Another way to remove cytokines is using specific HD membranes. Due to the huge size of cytokines as part of middle molecules, high cut-off membranes with a larger pore size (>0.01 µm) are used [46]. These membranes are intended to effectively remove molecules from 20 kDa to 50 kDa. It was clinically proven that high cut-off membranes could effectively remove cytokines, particularly IL-4, IL-6, IL-8, and IL-12, without eliminating albumin. The clearance of cytokines resulted in substantial enhancement of organ dysfunction and haemodynamic condition.

On the other hand, an increased level of endotoxins is usually observed in patients with AKI. Among the patients, endotoxins are found circulating in the bloodstream after encountering severe traumas. After a few times undergoing dialysis treatment, endotoxins originated from bacterial lipopolysaccharides (i.e., Gram-negative bacteria) found in the dialysate may accumulate in the blood due to their diffusion into the blood during the treatment. This would elevate the level of endotoxins in the patient’s blood. To remove the circulating endotoxins from the patient’s blood, Toray Medical Co. Ltd. (Tokyo, Japan) has developed Toraymyxin, a polymyxin B-immobilised fibre blood purification column. The product was approved by the Japanese National Health Insurance system to treat endotoxemia and septic shock. Polymyxin B is known for its bactericidal effect against Gram-negative bacteria. Besides, it can inhibit the effects of endotoxins via binding with the active site (lipid A domain) of the endotoxin molecules. For selective adsorption of endotoxins, polymyxin B is immobilised on the surface of polymeric fibres (membrane). Mortality risk studies on various targeted groups revealed that the use of Toraymyxin significantly reduced the mortality risk ratio [50].

Due to the serious threats imposed by cytokines and endotoxins, the focus is now on developing HD membranes that could remove these two types of molecules simultaneously from blood. oXiris membrane was developed recently, which comprised of AN69 membrane, surface-treated with polyethylenimine, and grafted with heparin on the inner membrane surface. AN69 membrane alone can remove large molecular weight molecules, including cytokines and uremic toxins, by membrane binding [57]. To tackle the problem of endotoxin accumulation, the membrane was surface-treated with polyethylenimine as this polymer can adsorb endotoxins. The permanent grafting of heparin on the membrane surface inhibited blood coagulation by adsorbing antithrombin III (ATIII) from blood, forming a stable ATIII-thrombin complex as high as ~270 ng/mL compared to ~10 ng/mL when using non-heparin-grafted AN69 membrane. The grafted heparin has been proposed to catalyse the conversion of ATIII into a potent anticoagulant in the bloodstream. The improved membrane thrombogenicity could minimise the problem of membrane clotting, which is the most frequent technical complication encountered during continuous renal replacement therapy for patients with AKI.

To date, oXiris is the only membrane (dialyser) with great removal of both endotoxins (68%) and cytokines (>90%) [58]. This membrane can function as a dialyser in HD or as an adsorbent in haemoperfusion. Clinical improvements, such as increased arterial pressure and reduced norepinephrine dose, have been observed following the use of oXiris membrane on patients with AKI. These lead to improved organ function and haemodynamic stability.

## 7. Combination of Haemodialysis and Haemoperfusion

Ghezi et al. [53] studied the combination of two dialysers in series, called paired filtration dialysis (PFD) technique or HDF mode. This technique has been developed due to the weakness of high-flux HD, which is the process interference between diffusion and convection as they occur simultaneously, thus decreasing their respective efficiency. In studying a series of dialysis modes, the following mathematical reasoning is used [59];
Coefficient of haemodiafiltration: K_hdf_ = K_uf_ + K_d_(1)
Coefficient of UF or convection: K_uf_ = Q_uf_ (C_o_/C_i_)(2)
However, C_o_ < C_i_, and therefore C_o_/C_i_ < 1.0(3)
Then, K_uf_ < Q_uf_(4)
Consequently, [K_d_ + K_uf_] < ([K_d_] + [K_uf_])(5)
where K_hdf_ is the coefficient of haemodiafiltration; K_uf_ is the coefficient of UF; K_d_ is the coefficient of diffusion; Q_uf_ is UF flow rate; C_o_ is concentration out; C_i_ is concentration in; [ ] is one chamber only; ([ ] + [ ]) is one dialyzer with two chambers.

Based on the above mathematical reasoning, two processes that occur simultaneously in a single dialyser are less efficient compared to when they occur in different dialysers. Based on this explanation, a combination of HD and haemoperfusion (Figure 1) as a function of filtration (diffusion and UF) and adsorption is an ingenious idea to remove a wide range of uremic toxins.

The combination of diffusion, UF, and adsorption using HD and haemoperfusion techniques could offer many advantages. Firstly, the fouling or blocking of pores in membrane surfaces can be avoided [40]. Secondly, middle molecules and protein-bound uremic toxins would be first adsorbed in the haemoperfusion system, leaving only small water-soluble molecules in the blood to be removed by dialysis (diffusion) through HD membrane. Under these arguments, it can be assumed and predicted that the combination of HD and haemoperfusion techniques promotes excellent uremic toxin clearance [36].

Based on the description above, dialysis treatments can be divided based on the principle and their capability to remove uremic toxins, as shown in Table 2. Based on the economics in terms of time efficiency and the adsorption principles of haemoperfusion, this treatment takes less time than other treatments. However, it should be stated that the selection of the type of dialysis treatment must be based on the patient’s needs. Based on the consideration of completeness of uremic toxins adequacy, the treatment using a combination of diffusion and adsorption is very promising in the future. It helps to improve the quality of life of kidney failure patients.

## 8. Conclusions

Improper selection of dialysis membranes and the modes of dialysis would result in the inadequacy of blood purification treatment, which will be very detrimental to the patients with AKI and can lead to death. The top priority of choosing a dialysis membrane must be based on its capacity to remove targeted groups of molecules according to the clinical profile of an individual rather than the cost of treatment so that an efficient blood purification process can be achieved. A low-flux membrane, for example, is sufficient to remove small water-soluble molecules over a long period of time for septic patients with unstable haemodynamic conditions. In a high concentration of middle molecules in the blood, a high-flux membrane is needed. High-flux membranes are more versatile as they can be utilised in HD, HF, and HDF modes.

On the other hand, adsorptive membranes have been developed to improve the removal of protein-bound uremic toxins and different types of molecules, including cytokines and endotoxins, through diffusion and adsorption. As an alternative, haemoperfusion treatment using a sorbent cartridge is performed to eliminate these recalcitrant molecules due to its higher efficiency compared to HD using an adsorptive membrane. The removal of cytokines and endotoxins from blood, either by adsorptive membrane or adsorbent, is very beneficial in improving the clinical outcomes of septic patients with AKI. In some cases, the integration of haemoperfusion and HD is made to systematically remove all types of uremic toxins at one time.

## Figures and Tables

**Figure 1 membranes-12-00325-f001:**
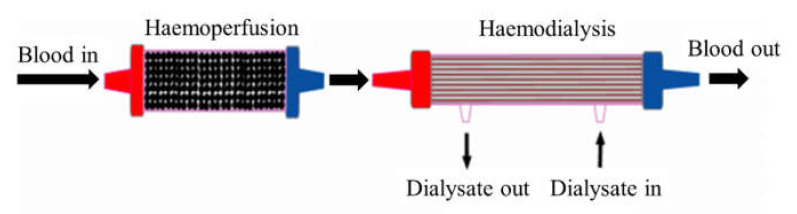
Integration of HD and haemoperfusion.

**Table 1 membranes-12-00325-t001:** Uremic toxins based on their physicochemical characteristic along with their molecular weight (MW).

Water-Soluble Low-Molecular-Weight	MW	Middle-Molecules	MW	Protein-Bound	MW
1-methyladenosine	281	Adrenomedullin	5729	2-methoxyresorcinol	140
1-methylguanosine	297	Atrial natriuretic peptide	3080	3-deoxyglucosone	162
1-methylinosine	282	β_2_-microglobulin	11,818	3-carboxyl-4-methyl-5-propyl-2-furanpropionic acid	240
Asymmetrical dinethylarginine	202	β-endorphin	3465	Fructoselysine	308
α-keto-δ-guanidinovaleric adic	151	Cholecystokinin	3866	Glyoxal	58
α-N-acetylarginine	216	Clara cell protein	15,800	Hippuric acid	179
Arab(in)itol	152	Complement factor D	23,750	Homocysteine	135
Arginnic acid	175	Cystatin C	13,300	Hydroquinone	110
Benzylalcohol	108	Degranulation inhibiting protein	14,100	Indole-3-acetic acid	175
β-guanidinopropionic acid	131	Delta-sleep inducing peptide	848	Indoxyl sulphate	251
β-lipoprotin	461	Endothelin	4283	Kynurenine	208
Creatine	131	Hyaluronic acid	25,000	Kynurenic acid	189
Creatinine	113	Interleukin-1β	32,000	Leptin	16,000
Cytidine	234	Interleukin-6	24,500	Melatonin	126
Dimethylglycine	103	κ-Ig light chain	25,000	Methylglyoxal	72
Erythritol	122	λ-Ig light chain	25,000	N^ε^-(carboxymethyl)lysine	204
γ-guanidinobutyric acid	145	Leptin	16,000	*p*-cresol	108
Guanidine	59	Methionine-enkephalin	555	Pentosidine	342
Guanidinoacetic acid	117	Neuropeptide	4272	Phenol	94
Guanidinosuccinic acid	175	Parathyroid hormone	9225	P-OH hippuric acid	195
Hypoxanthine	136	Retinol-binding protein	21,200	Putrescine	88
Malondialdehyde	71	Tumor necrosis factor-α	26,000	Quinolinic acid	167
Mannitol	182			Retinol-binding protein	21,200
Methyguanidine	73			Spermidine	145
Myoinositol	180			Spermine	202
N^2^,N^2^-dimethylguanosine	311				
N^4^-acetylcytidine	285				
N^6^-methyladenosine	281				
N^6^-threonylcarbamoyladenosine	378				
Orotic acid	174				
Orotidine	288				
Oxalate	90				
Phenylacetylgluatmine	264				
Pseudouridine	244				
Symmetrical dimethylarginine	202				
Sorbitol	182				
Taurocyamine	174				
Threitol	122				
Thymine	126				
Uracil	112				
Urea	60				
Uric acid	168				
Uridine	244				
Xanthine	152				
Xanthosine	284				

**Table 2 membranes-12-00325-t002:** Types of dialysis treatment based on the principle and their capability to remove the uremic toxins.

Basic Principle	Treatment/Membrane Type	Uremic Toxins Removed
Diffusion	Dialysis/Low-flux membrane	Water-soluble
	Dialysis/High-flux membrane	Middle-molecules and Protein-bound
Adsorption	Haemoperfusion	Protein-bound
Combination of diffusion and adsorption	Dialysis/Mixed Matrix Membrane (MMM)	Water-soluble, middle-molecules, and protein-bound

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
