# Peer review of "Dialysis Membranes for Acute Kidney Injury"

_membranes, 2022, doi:10.3390/membranes12030325_

Round 1
Reviewer 1 Report
This paper summarizes methods and progress towards helping patients diagnosed with AKI which is an important topic worth reviewing and I appreciate the authors effort in this area. As written however, I do not feel it would be appropriate to publish in Membranes. With extensive revisions I do believe that this work could become suitable for publication following additional review. Below are more detailed comments.
- This manuscript requires a significant amount of additional editing to improve sentence structure, grammar, readability, typos, etc. As currently written, it is difficult for me to fully review some portions due not having full confidence in my interpretation of what is being said. This is true throughout the manuscript, however it is most significant in the Introduction. For example:
- Page 2: As written it is not clear if only prerenal and postrenal AKI can be treated in 7 days, or if that also includes renal. Farther down the same page, it is stated that AKI is defined as occurring within 48 hours, but earlier it is said less than 7 days. Additional clarity is needed on which definitions apply to which situations and the wording used to describe things.
- There are multiple cases of abbreviations not being defined prior to first use, examples include:
- Page 1: HD is not defined until later
- Page 10: MMM is not defined, AC also not defined
- Page 11 HAP not defined
- The paper would benefit from additional citations in order to help answer questions related to the relevant reported claims. Here are two examples:
- Page 2: the number of AKI cases per million people per year is given. Is this worldwide or in a specific population? Which year? Etc.
- Page 2: 30-80% of people suffer from AKI and ACKF. This is quite a large range, why? Where is it taken from? Is this referring to AKI and ACKF combined?
- Page 9: What is meant by “high quality” dialysate
- The value of this review would be greatly improved by discussion on more recent developments, if available. Very few of the discussed and cited works are from the last 5 years. I would encourage the authors to include any recent relevant advancements, or if none are available, discuss why.
- Some citations are not properly formatted or are incomplete.
Author Response
Dear Reviewer,
Thank you very much for the constructive advice to improve the quality of our paper. Please see the attachments for our answer for your comments and suggestions. We also attached our revised manuscript. On the revised manuscript, we write on the red color for additional descriptions based on the
reviewer comments and blue color improvement in English language/
proofread that We have done.
Thank you.

Reviewer 2 Report
Please read my details review and revision your manuscript. Thank you

Author Response
Dear Reviewer,
Thank you very much for the constructive advice to improve the quality of our paper. Please see the attachments for our answer for your comments and suggestions. We also attached our revised manuscript. On the revised manuscript, we write on the red color for additional descriptions based on the
reviewer comments and blue color improvement in English language/
proofread that have been done.
Thank you.

Round 2
Reviewer 1 Report
The authors have made significant changes to improve the manuscript and address the majority of my comments. I believe that with minor revisions the text could be appropriate for publication. I have a few additional comments/responses below.
1) The readability of the manuscript is significantly improved to the point where it is much easier to understand the meaning of the text. There are still, however, sufficient grammatical mistakes and areas in which clarity could be improved. The text would greatly benefit from another round of editing.
2) I appreciate the authors clarification of the definition of AKI in point 1, however, I still feel that additional changes would be beneficial. It seems there are different standards to diagnose AKI and the citation given in the text [5] defines one criterion as a >50% increase in creatinine within 48 hours (not 7 days). If instead the authors are referring to guidelines from KDIGO, then the definition would be >50% within 7 days. [Kidney Disease: Improving Global Outcomes (KDIGO). Acute Kidney Injury Work Group. KDIGO clinical practice guidelines for acute kidney injury. Kidney Int Suppl 2012; 2:1.]. I believe this issue could be clarified by providing a citation for the initial statement referencing 7 days and making it clear if this comes from a different place than the later statement referencing 48 hours. Additionally, citation [5] defines an AKI diagnosis as a change in creatinine levels within 48 hours, not sustained over 48 hours. As a result, the text “AKI is a sudden reduction of kidney function for 48 hours” may be confusing. Should it instead read “AKI is a sudden reduction of kidney function within 48 hours”?
3) In response to point 3, it was stated that the statistic stating 30-80% of people suffering from kidney failure has been removed. It appears to still be present in the text (although now more broadly referencing kidney disease worldwide instead of AKI and ACKF). This claim is still not cited, and it is unclear what it means. Why such a large range? What does this range represent and how does it relate to the above statement specific to the USA?
4) The addition of a definition for high quality dialysate solution is appreciated. The citation, however, appears to be in an incorrect format in the text.
Author Response
Dear Professor/Dr.
Thank you very much for the valuable comments and suggestions. Please see the attachment for our clarification.
Thank you.
